# Characterisation of physicochemical parameters and antibacterial properties of New Caledonian honeys

Marcela Bucekova[1], Jana Godocikova[1], Romain Gueyte[2], Céline Chambrey[2], Juraj Majtan [1,3]*

**1** Laboratory of Apidology and Apitherapy, Department of Microbial Genetics, Institute of Molecular Biology, Slovak Academy of Sciences, Bratislava, Slovakia, **2** Beekeeping Center, ADECAL Technopole, Noumea Cedex, New Caledonia, **3** Department of Microbiology, Faculty of Medicine, Slovak Medical University, Bratislava, Slovakia

* juraj.majtan@savba.sk

**Data Availability Statement:** All relevant data are within the paper.

**Funding:** This work was supported by the Scientific Grant Agency of the Ministry of Education of the

## Abstract

Honey is an attractive natural product with various health benefits. A few honey-based commercial products have successfully been adopted in clinics to improve wound healing. However, screening of other potential sources of medical-grade honey, in particular, honeys from territories with high floral species diversity and high endemicity, is highly needed. The goal of this study was to characterise the physicochemical and antibacterial properties of New Caledonian honey samples (n = 33) and to elucidate the major mechanism of their antibacterial action. Inhibitory antibacterial activity of honeys against *Staphylococcus aureus* and *Pseudomonas aeruginosa* was determined with a minimum inhibitory concentration (MIC) assay. Enzymatic activity of glucose oxidase and the content of hydrogen peroxide ($H_2O_2$) in honey samples were analysed. Furthermore, total protein content of honeys together with their electrophoretic protein profiles were also determined in the study. The antibacterial efficacy of 24% of the tested honey samples was slightly superior to that of manuka honey with unique manuka factor 15+. The antibacterial activity of catalase-treated honey sample solutions was significantly reduced, suggesting that $H_2O_2$ is a key antibacterial compound of diluted honeys. However, the kinetic profiles of $H_2O_2$ production in most potent honeys at a MIC value of 6% was not uniform. Under the experimental conditions, we found that a $H_2O_2$ concentration of 150 μM in diluted honeys is a critical concentration for inhibiting the growth of *S. aureus*. In contrast, 150 μM $H_2O_2$ in artificial honey solution was not able to inhibit bacterial growth, suggesting a role of phytochemicals in the antibacterial activity of natural honey. In addition, the continuous generation of $H_2O_2$ in diluted honey demonstrated an ability to counteract additional bacteria in re-inoculation experiments. In conclusion, the tested New Caledonian honey samples showed strong antibacterial activity, primarily based on $H_2O_2$ action, and therefore represent a suitable source for medical-grade honey.

Slovak Republic and the Slovak Academy of Sciences VEGA 2/0022/22 and the Slovak Research and Development Agency under Contract No. APVV-21-0262. The funders had no role in study design, data collection and analysis, decision to publish, or preparation of the manuscript.

**Competing interests:** The authors have declared that no competing interests exist.

## Introduction

Antimicrobial drug resistance represents a major burden on global health and jeopardises the effectiveness of currently used antimicrobials. It is therefore crucial to reduce and optimise the use of antibiotics and introduce antimicrobial stewardship programmes to tackle drug-resistant infections globally [1]. Although a recent systematic review and meta-analysis showed promising results achieved with such programmes [1], it seems that the problem of antimicrobial drug resistance is more complex and linked to climate change (e.g., higher concentrations of contaminants in the environment) [2]. In addition, the use of topical antibiotics is still subject of debate because of their overall efficiency and the development of bacterial resistance [3]. In the last decade, natural products have become attractive platform for developing new antimicrobials which might help to tackle the problem of bacterial resistance [4]. Indeed, natural products are diverse and rich sources of antimicrobial compounds. Marine [5], insect [6] and plant [7] products are widely studied for their multi-factorial antimicrobial activities and perspectives to identify novel antimicrobial drugs which are effective against multi drug-resistant microorganisms. However, most of these natural products and their isolated antimicrobial compounds are mainly implicated for topical therapy because their minimal inhibitory concentrations (MICs) are substantially higher than the possible systemic concentrations achieved in oral therapies [8]. Topical antimicrobials are often used for the elimination of wound infections, which are considered major factors responsible for the delay in wound healing.

Nowadays, natural products such as honey and essential oils are attractive antibacterial agents in the management of infected acute and chronic wounds [9, 10]. Honey has been used in clinical practice for decades and currently, several medical-grade honey-based products are available as medical devices worldwide [11]. Interestingly, honey can effectively be used together with other topically applied antiseptics (*e.g.*, benzalkonium bromide, silver (I) nitrate or tea tree oil) for treating superficial bacterial infections [12, 13].

Manuka monofloral honey, made from the nectar of the manuka bush (*Leptospermum scoparium*), which is native to New Zealand and Australia, is unique among other honeys due to its higher content of methylglyoxal (MGO). This compound is the key component responsible for the non-peroxide antibacterial activity exerted by manuka honey [14]. High amounts of MGO are present in manuka honey, even more than 1,000 mg/kg, up to 100-fold higher compared to those found in most other honeys [15, 16]. Globally, manuka honey is the most frequently used honey in wound care.

Over the past decade, intense screenings of other honey types have been performed to identify new sources of medical-grade honey that would be attractive for the pharmaceutical industry [11]. In many cases, the antibacterial activity of newly described honey types is often compared to that of manuka honey, with different levels of MGO or different unique manuka factor (UMF) [17–25]. However, the mechanism of antibacterial action in all analysed non-manuka honeys is based on the action of hydrogen peroxide ($H_2O_2$). The generation of $H_2O_2$ in diluted honey is mainly mediated through the action of bee-derived enzyme glucose oxidase (GOX) [26]. However, the level of accumulated $H_2O_2$ may be significantly diverse among honeys, depending on several factors [27]. Interestingly, honeydew honeys contain significantly higher amounts of generated $H_2O_2$ compared to blossom honeys, probably due to the considerable presence of polyphenolic compounds [20, 25]. As stated above, $H_2O_2$ is produced mainly enzymatically and its amount produced in honey significantly correlates with honey minimum inhibitory concentration (MIC) [27]. On the other hand, some studies have reported a lack of correlation between the $H_2O_2$ level and overall antibacterial activity [25, 28]. Therefore, it is required to elucidate observed discrepancies by determining the critical concentration of accumulated $H_2O_2$ and identifying phytochemicals which can enhance honey antibacterial effect.

Additional screening the antibacterial potential of traditional botanical honey types such as acacia, linden, buckwheat, clover, sunflower and other honeys, despite their different geographical origins, did not very likely identify different mechanisms of antibacterial action. Therefore, it is important to focus on territories with high floral species diversity and high endemicity to identify honey samples with high antibacterial efficacy and possible with different mechanisms of antibacterial action. One of such territories is New Caledonia, which is characterised by a remarkably high species diversity as well as due to their high endemicity and an unusual abundance of archaic plant taxa [29]. New Caledonia is remarkable for its high rate of endemism; 75% (around 2500 species) of the vascular plant species are found nowhere else in the world [30, 31]. The flora of New Caledonia has links with Australia, New Guinea and New Zealand [30].

In this study, we characterised a representative collection of New Caledonian honey samples (n = 33) in terms of physicochemical and antibacterial properties. Furthermore, we analysed the protein content and enzymatic activity of GOX in honey samples. We also described the mechanism of the antibacterial action of the tested honeys at their MICs and elucidated the role of accumulated $H_2O_2$ in inhibiting the growth of *Staphylococcus aureus* (*S. aureus*).

## Materials and methods

### Honey samples

A total of 33 honey samples from local beekeepers from all regions of New Caledonia were evaluated. The samples were harvested in 2021 and identification of the floral source of each honey sample using mellisopalynological analysis according to Erdtman [32] was performed by CARI asbl, Louvain-La-Neuve, Belgium. Two different commercial manuka honeys, Comvita UMF15+ (Comvita NZ, Ltd., New Zealand) and Manuka Honey MGO 550+ (Manuka Health New Zealand, Ltd., New Zealand) were also evaluated. All collected samples were kept at 4°C in the dark.

As a negative control artificial honey (AH) was prepared by dissolving 39 g d-fructose, 31 g d-glucose, 8 g maltose, 3 g sucrose and 19 g distilled water, as described elsewhere [33].

### Bacteria

The tested bacterial isolates *Staphylococcus aureus* CCM4223 (Czech Collection of Microorganisms, Brno, Czech Republic) and *Pseudomonas aeruginosa* CCM1960 (Czech Collection of Microorganisms, Brno, Czech Republic) were acquired from the Department of Medical Microbiology at the Slovak Medical University in Bratislava, Slovakia. Both bacteria were stored at -80°C in Muller-Hinton broth (MHB) /glycerol stocks. Bacterial working cultures were cultured on MHB agar then stored at 4°C and were refreshed from frozen stocks every two weeks.

### Physicochemical and organoleptic characteristics of honey

All physicochemical analyses were performed at CARI asbl, Louvain-La-Neuve, Belgium. Analytical methods measuring pH, electric conductivity, hydroxymethylfurfural (HMF) and the fructose/glucose (F/G) ratio were performed according to "Harmonized Methods of the International Honey Commission" (2009) and the European Directive 2001/110/EC and humidity according to APAQ-W and German guidelines for honey (Leitsätze) [34, 35].

Aliquots of honey samples were heated to 50°C to dissolve sugar crystals, and the colour was determined by the spectrophotometric (Genesys 10 UV, Thermo Scientific, UK) measurement of the absorbance of a 50% honey solution (w/v) at 635 nm. The honeys were classified

according to the Pfund scale after conversion of the absorbance values: mm Pfund $= -38.70 + 371.39 \times$ Abs [36].

## Determining the protein profile of honey samples and total protein content

For protein determination, 15 µL aliquots of diluted honey samples (50% w/w in distilled water) were loaded on 12% SDS-PAGE electrophoresis gels and separated using a Mini-Protean II electrophoresis cell (Bio-Rad, Hercules, CA, USA).

The total protein content (TPC) was measured using the Quick Start™ Bradford reagent (Bio-Rad, Hercules, CA, USA), according to the manufacturer's instructions.

## Determination of GOX enzymatic activity

The bee-derived GOX activity was determined with a Megazyme GOX assay kit (Megazyme International Ireland Ltd, Bray, Ireland), which is based on the oxidative catalysis of β-D-glucose to D-glucono-δ-lactone, with the concurrent release of $H_2O_2$. The resultant $H_2O_2$ reacts with p-hydroxybenzoic acid and 4-aminoantipyrine in the presence of peroxidase to form a quinoneimine dye complex, which has a strong absorbance at 510 nm. The enzyme activity was determined in freshly prepared and centrifuged (10,000 $g$, 5 min) 20% (w/v) honey solutions in a 96-well microplate, according to the manufacturer's instructions.

## Determination of the $H_2O_2$ concentration after 24h and $H_2O_2$ kinetics

The $H_2O_2$ content in the honey samples was determined using a Megazyme GOX assay kit (Megazyme International Ireland Ltd); based on the release of $H_2O_2$. As a standard, 9.8–312.5 µM diluted $H_2O_2$ was used. 40% (w/w) of the honey solutions in 0.1 M potassium phosphate buffer (pH 7.0) were prepared. After 24 h of incubation at 37˚C, each honey sample and $H_2O_2$ standard were tested in duplicate in a 96-well microplate. Absorbance was measured at 510 nm using a Synergy HT microplate reader (BioTek Instruments, VT, USA).

In order to mimic the experimental conditions during determination of antibacterial activity of honey samples, $H_2O_2$ production was also measured in selected honey solutions which were diluted with MHB instead of phosphate buffer (as stated above) to a final honey concentration corresponded to their MIC values. The level of $H_2O_2$ was determined in honey solutions after incubation at 37˚C at 0, 1, 2, 4, 6, 8 and 24 h using a Megazyme GOX assay kit.

## Determining the antibacterial activity of honey

The honey samples were subjected to an antibacterial MIC assay to determine their antibacterial activity against *S. aureus*, following the modified method of Bucekova et al. [26] based on broth microdilution method described by the Clinical and Laboratory Standards Institute [37]. Bacteria were cultured in MHB at 37˚C overnight. Bacterial cultures were suspended in phosphate-buffered saline (PBS), with a pH of 7.2, and the turbidity of the suspension was adjusted to $10^8$ colony-forming units (CFU)/mL and diluted with MHB medium (pH 7.3 ± 0.1) to a final concentration of $10^6$ CFU/mL. The final volume in each well of sterile 96-well polystyrene U-shaped plates (Sarstedt, Germany) was 100 µL, consisting of 90 µL of sterile MHB medium (as a positive control) or diluted honey sample and 10 µL of bacterial suspension. Each honey sample dilution was prepared from a 50% honey solution (w/w in MHB medium) by further dilution with the MHB medium, resulting in final concentrations of 40%, 35%, 30%, 25%, 20%, 18%, 16%, 14%, 12%, 10%, 8%, 6% and 4%. Wells containing 100 µL of sterile MHB medium was considered as a negative control. After 18 h of incubation at 37˚C and 1,250 rpm, the inhibition of bacterial growth was determined visually as the lowest concentration of

honey completely inhibiting bacterial growth, resulting in an optically clear well (lack of visual turbidity) and expressed as a MIC value. All tests were performed in triplicate and repeated three times.

Bactericidal activity, expressed as minimal bactericidal concentration (MBC) of honey solution, was evaluated in selected honey samples No. 1, 3, 6, 7, 8, 11, 15 and 27, according to Bucekova et al (2019) [26] with some modifications Briefly, the viability of bacteria in wells with no visible bacterial growth (wells around MIC) was determined by spotted aliquots (10 μL) from each well onto MHB agar plates and incubated overnight at 37˚C. The highest honey dilution at which all bacterial cells were killed was recorded as the MBC.

In addition, the viability of bacteria in honey solutions at different MBCs was determined in first 3 hours after bacterial inoculation. Aliquots (10 μL) from each timepoints (0, 1, 2 and 3 h) were subsequently spotted on MHB agar and incubated overnight at 37˚C. Colony-forming units were counted and expressed as CFU/μL. As control AH, AH with 300 μM $H_2O_2$ and MHB alone were used.

## Enzymatic treatment of honey samples with catalase

The selected honey samples No. 1, 3, 6, 7, 8, 11 and 15, at a concentration of 50% (w/w), were treated with catalase (2,000–5,000 U/mg protein; Sigma-Aldrich, UK) at a final concentration ranging from 1,000–2,500 U/ml at ambient temperature for 2 h. Catalase-treated honey samples were then used in the antibacterial assay to determine the MIC values against *S. aureus*.

## Statistical analysis

The physicochemical parameters of the honey samples were expressed as mean with standard uncertainty ($k = 2$). The Shapiro-Wilk test of normality was used to determine the data distribution. Analysis of variance (ANOVA) with Tukey's multiple comparison test was used to determine differences among bactericidal activities. Values with $P$ below 0.05 were considered statistically significant. A principal components analysis (PCA) was conducted to examine the relationship among selected honey physicochemical parameters, botanical origin, activity of GOX and antibacterial activity. All statistical analyses were performed using GraphPad Prism version 9.2.0 (GraphPad Software Inc., La Jolla, CA, USA).

## Results

### Melissopalynological analysis

Overall, 17 botanical families were identified in honey samples; 21.2% of the samples contained pollen from one botanical family, 33.3% contained pollen from two families, and 45.5% of the samples contained mixed pollen from more than three families (Table 1 and S1 Table). The most dominant family was Anacardiaceae, present in 51.5% of the samples, followed by Myrtaceae together with Cunoniacea, found in 48.5% and 45.5% of the samples, respectively. Qualitative pollen analysis highlighted seven samples (No. 1, 3, 29, 27, 20, 10 and 18) containing only one type of pollen, with the following frequencies: Myrtaceae 89% and 72%, Cunoniacea 80%, 70%, 59%, Fagaceae 81% and Mimosaceae 85%, respectively. Moreover, Myrtaceae pollen was predominant also in samples No. 6, 8 and 32. The family Anacardiaceae was predominant in samples No. 25, 26, 12 and 4 with 70%, 61%, 46% and 45%, respectively. Cunoniacea was present in samples No. 29, 27, 20 and 13, with a pollen coverage of 80%, 70%, 59% and 58%, respectively. The lowest amount of detected pollen was observed for Zygophyllaceae, which was found in two samples (No. 31 and 32), accounting for 11% and 12%, and for Rosaceae, which was only found in one sample (No. 30), with a pollen coverage of 14%.

**Table 1. Melissopalynological analysis of collected New Caledonian honey samples (n = 33).**

| Family | Max. (%) | ≥46% | 45–16% | 15–4% | Present |
|---|---|---|---|---|---|
| Anacardiaceae | 70 | 3 | 10 | 4 | 17 |
| Apiaceae | 35 | - | 3 | - | 3 |
| Apocynaceae | 46 | 1 | 2 | - | 3 |
| Brassicaceae | 16 | - | 1 | - | 1 |
| Casuarinaceae | 30 | - | 1 | - | 1 |
| Cunoniaceae | 80 | 4 | 8 | 3 | 15 |
| Euphorbiaceae | 31 | - | 2 | - | 2 |
| Fagaceae | 81 | 1 | 1 | - | 2 |
| Mimosaceae | 85 | 3 | 5 | 2 | 10 |
| Myrtaceae | 89 | 5 | 9 | 2 | 16 |
| Poaceae | 22 | - | 1 | - | 1 |
| Rhizophoraceae | 40 | - | 1 | - | 1 |
| Rosaceae | 14 | - | - | 1 | 1 |
| Salicaceae | 28 | - | 1 | - | 1 |
| Zygophyllaceae | 12 | - | - | 1 | 1 |

Max.: maximum value reached by the pollen type in samples, present: number of samples where the pollen type was identified. t.: pollen type common for different plant genera.

## Physicochemical parameters of honey samples

The basic physicochemical parameters of New Caledonian honey samples are listed in Table 2. Two samples, No. 18 and 30, did not meet the Codex standard criterium for the HMF level (< 80 mg/100 g for tropical honeys) and moisture (< 20%), respectively.

In total, 8 out of 33 honey samples had an electric conductivity value above 0.8 mS/cm. In the collected honey samples, the pH values ranged from 3.9 to 5.1.

## Protein content, GOX enzymatic activity and $H_2O_2$ accumulation in diluted honey samples

The TPC and an SDS-PAGE profile of each New Caledonian honey sample were determined (Table 3 and S1 Fig). As shown in Table 3, the mean TPC values substantially varied among samples and ranged from 448 to 1,282 µg/g of honey. The SDS-PAGE profiles of honey sample solutions were comparable with that of major royal jelly protein 1 (MRJP1) as a dominant honey protein. There was only a slight shift of MRJP1 protein in samples No. 4, 5 and 10 (S1 Fig).

The enzymatic activity of GOX was evaluated in all honey samples (Table 3), resulting in great differences in enzymatic activity among the tested samples. The lowest enzymatic activity was determined in sample No. 21 (calciferous forest), with a value of 0.10 ± 0.18 mU/mL, and the highest activity was documented in sample No. 15 (anthropized environment), with value of 13.34 ± 2.80 mU/mL. Statistical analysis showed no correlation ($r_s$ = -0.072, P = 0.689) between the enzymatic activity of GOX and TPC in the analysed samples.

Similar to the enzymatic activity of GOX, there was a great variance in the concentration of $H_2O_2$, determined in 40% honey solutions after incubation for 24 h, in the analysed honey samples (Table 3), ranging from 285 to 4,057 µM. Due to large variations within the measured parameters, statistical analysis did not reveal any correlation ($r_s$ = -0.048, P = 0.790) between the level of $H_2O_2$ and the enzymatic activity of GOX in the analysed samples.

**Table 2. Physicochemical and organoleptic characteristics of New Caledonian honey samples (n = 33).**

| Sample No. | Humidity (%) | pH | EC (mS/cm) | HMF (mg/100 g) ± SD | F/G | Pfund |
|---|---|---|---|---|---|---|
| 1 | 16.44 | 4.7 | 1.267 | 6.3±2.4 | 1.39 | 5.50 |
| 2 | 18.32 | 4.2 | 1.235 | 30.1±5.1 | 1.42 | 72.72 |
| 3 | 16.55 | 4.3 | 1.306 | 12.4±2.4 | 1.30 | 11.44 |
| 4 | 16.39 | 4.5 | 0.643 | 26.7±4.5 | 1.73 | 98.71 |
| 5 | 17.23 | 4.5 | 0.604 | 20.9±3.6 | 1.42 | 76.06 |
| 6 | 16.13 | 4.8 | 1.464 | 1.5±2.4 | 1.44 | -3.05 |
| 7 | 17.33 | 4.3 | 0.796 | 11.2±2.4 | 1.16 | 119.14 |
| 8 | 16.73 | 4.5 | 1.234 | 16.5±2.8 | 1.40 | 10.69 |
| 9 | 18.6 | 4.2 | 0.970 | 32.1±5.5 | 1.34 | 70.86 |
| 10 | 19.26 | 4.4 | 0.611 | 14.8±2.5 | 1.63 | 66.03 |
| 11 | 18.43 | 3.9 | 0.558 | 19.0±3.2 | 1.13 | 28.89 |
| 12 | 14.81 | 5.0 | 0.772 | 3.6±2.4 | 1.80 | 64.92 |
| 13 | 15.61 | 4.9 | 0.810 | 5.0±2.4 | 1.67 | 54.15 |
| 14 | 16.24 | 4.8 | 0.912 | 5.8±2.4 | 1.79 | 56.00 |
| 15 | 17.14 | 4.3 | 0.752 | 1.8±2.4 | 1.24 | 21.84 |
| 16 | 16.46 | 4.2 | 0.686 | 7.8±2.4 | 1.23 | 79.03 |
| 17 | 18.24 | 4.0 | 0.578 | 52.7±9.0 | 1.31 | 94.63 |
| 18 | 17.34 | 3.9 | 0.745 | 86.9±14.8 | 1.31 | 87.20 |
| 19 | 18.74 | 4.1 | 0.794 | 17.2±2.9 | 1.28 | 44.86 |
| 20 | 17.22 | 4.1 | 0.546 | 7.8±2.4 | 1.38 | 10.69 |
| 21 | 16.16 | 4.8 | 0.713 | 6.2±2.4 | 1.61 | 63.80 |
| 22 | 17.48 | 4.4 | 0.653 | 34.4±5.8 | 1.47 | 84.23 |
| 23 | 18.28 | 4.2 | 0.570 | 7.0±2.4 | 1.23 | 79.77 |
| 24 | 18.39 | 4.1 | 0.636 | 13.2±2.4 | 1.18 | 70.49 |
| 25 | 16.06 | 4.9 | 0.784 | 2.4±2.4 | 1.45 | 79.77 |
| 26 | 15.13 | 4.8 | 0.720 | 8.9±2.4 | 1.56 | 79.77 |
| 27 | 15.43 | 5.1 | 0.918 | 3.4±2.4 | 1.83 | 43.01 |
| 28 | 15.91 | 4.5 | 0.726 | 21.1±3.6 | 1.57 | 109.86 |
| 29 | 16.44 | 4.4 | 0.628 | 43.5±7.4 | 1.59 | 91.29 |
| 30 | 20.91 | 4.1 | 0.815 | 5.1±2.4 | 1.32 | 125.08 |
| 31 | 17.79 | 4.1 | 0.660 | 21.5±3.6 | 1.23 | 89.06 |
| 32 | 17.79 | 4.1 | 0.660 | 21.5±3.6 | 1.23 | 57.49 |
| 33 | 18.16 | 4.1 | 0.404 | 8.3±2.4 | 1.25 | 95.74 |

## Antibacterial effects of honey samples against *S. aureus* and *P. aeruginosa*

The mean MIC values of the tested honeys are shown in Fig 1 and ranged from 5.1% to 25% and from 6% to 18% for *S. aureus* and *P. aeruginosa*, respectively. Among the tested honeys, samples No. 4, 17 and 20 showed the lowest antibacterial efficacy against both bacterial species. However, none of the honey samples exhibited an antibacterial activity identical with that of artificial honey. Both manuka honey samples exhibited an equal antibacterial activity, with MIC values of 6%and 12% against *S. aureus* and *P. aeruginosa*, respectively. The antibacterial effects of eight (24% of honey samples) out of 33 New Caledonian honey samples were slightly superior to those of both manuka honeys. The most potent sample out of all tested honey samples was sample No. 15, with MIC values of 5.1% and 6.0% against *S. aureus* and *P. aeruginosa*, respectively.

**Table 3. Selected biochemical characteristics of New Caledonian honey samples (n = 33).**

| Sample No. | TPC (μg/g honey) ± SD | GOX (mU/mL) ± SD | $H_2O_2$ (μM) ± SD |
|---|---|---|---|
| 1 | 572.08±30.18 | 5.70±1.35 | 285.14±30.43 |
| 2 | 720.00±59.16 | 1.90±0.66 | 911.50±84.88 |
| 3 | 510.28±70.29 | 7.21±0.85 | 315.43±30.42 |
| 4 | 676.56±63.01 | 0.34±0.27 | 332.57±49.97 |
| 5 | 941.49±76.25 | 0.69±0.35 | 1252.00±28.05 |
| 6 | 785.79±133.73 | 11.67±2.42 | 306.29±32.72 |
| 7 | 1122.61±53.82 | 9.20±0.82 | 2607.43±41.11 |
| 8 | 839.01±49.73 | 6.11±0.27 | 366.29±60.80 |
| 9 | 804.64±31.34 | 5.42±0.59 | 1178.00±85.83 |
| 10 | 955.87±74.48 | 0.19±0.15 | 2059.00±108.71 |
| 11 | 887.13±43.05 | 8.08±1.48 | 2052.00±66.63 |
| 12 | 926.70±53.81 | 2.76±0.96 | 1136.50±34.46 |
| 13 | 933.58±75.85 | 2.05±0.29 | 1877.00±60.85 |
| 14 | 955.02±70.28 | 3.78±1.28 | 1060.50±70.40 |
| 15 | 1282.43±78.84 | 13.34±2.80 | 4056.73±452.45 |
| 16 | 723.92±48.80 | 9.71±1.48 | 3394.67±80.24 |
| 17 | 682.81±56.53 | 4.95±0.52 | 256.00±25.66 |
| 18 | 616.62±48.55 | 4.22±0.72 | 761.07±108.26 |
| 19 | 643.12±18.54 | 2.60±0.85 | 352.67±58.60 |
| 20 | 448.62±51.25 | 7.19±1.03 | 174.40±18.96 |
| 21 | 722.46±49.61 | 0.10±0.18 | 1256.00±28.80 |
| 22 | 1018.25±46.10 | 0.90±0.65 | 1767.00±67.30 |
| 23 | 647.47±51.42 | 6.39±0.94 | 2782.67±106.63 |
| 24 | 665.86±14.94 | 5.50±1.03 | 1184.57±130.13 |
| 25 | 607.80±18.64 | 2.26±0.24 | 1816.53±52.73 |
| 26 | 876.28±22.61 | 0.77±0.27 | 1422.29±46.80 |
| 27 | 670.03±65.17 | 3.72±1.36 | 1978.67±67.50 |
| 28 | 1227.25±41.63 | 2.78±1.02 | 710.50±45.32 |
| 29 | 1012.05±42.96 | 2.24±0.86 | 754.00±46.83 |
| 30 | 681.23±60.84 | 7.12±0.40 | 434.86±31.16 |
| 31 | 632.74±36.65 | 2.72±1.11 | 2284.00±86.73 |
| 32 | 556.26±89.86 | 1.95±1.05 | 1944.00±80.91 |
| 33 | 662.01±20.98 | 1.64±0.59 | 1553.00±76.50 |

TPC, total protein content; GOX, glucose oxidase

## Multifactorial analysis of selected physicochemical parameters, enzymatic activity of GOX, botanical origin and antibacterial activity of honey

According to PCA, the first two components (PC1 and PC2) explained 66.9% of the total variation in the selected physicochemical parameters (moisture, pH, HMF and electric conductivity), the enzymatic activity of GOX, the honey botanical origin and the antibacterial activity of honey against *S. aureus* and *P. aeruginosa* (Fig 2). A strong negative correlation was found between the enzymatic activity of GOX and the antibacterial activity of honey samples, expressed as MIC value, against *S. aureus* ($r_s$ = -0.572, $P < 0.001$) and *P. aeruginosa* ($r_s$ = -0.653, $P < 0.001$).

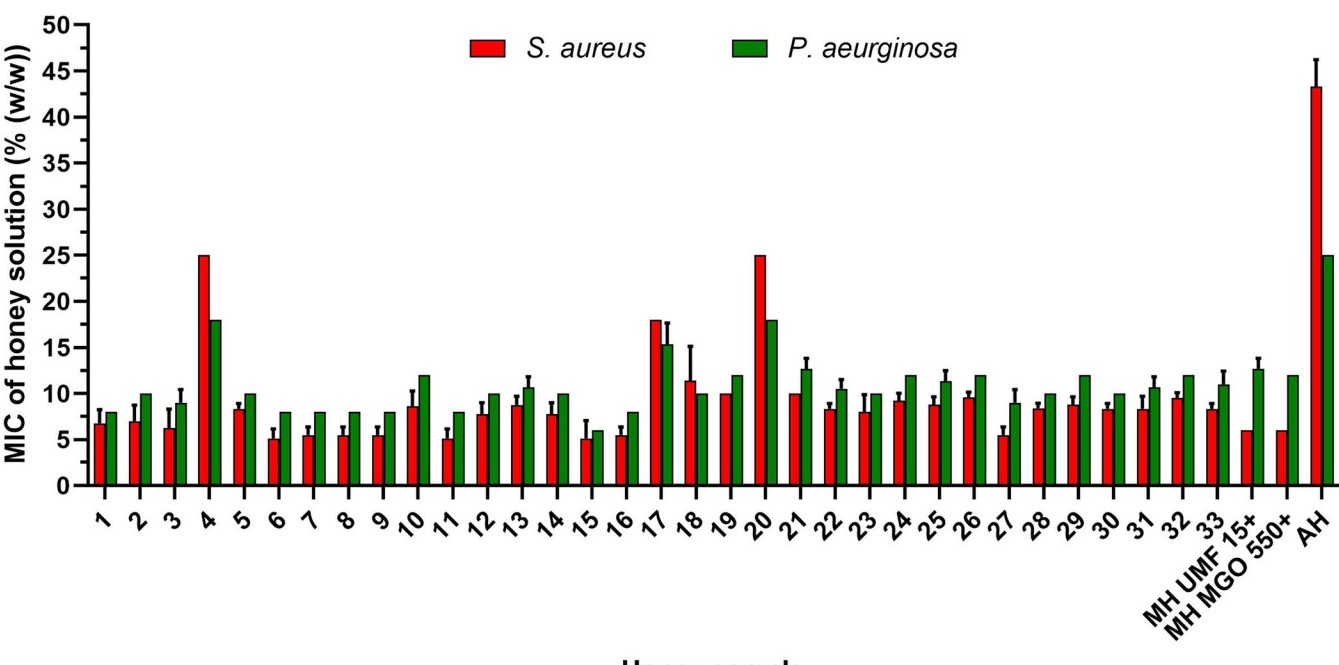

**Fig 1. Antibacterial activity of New Caledonian (n = 33) and manuka (UMF 15+ and MGO 550+) honey samples.** The antibacterial activity against *Staphylococcus aureus* and *Pseudomonas aeruginosa* was determined by a minimum inhibitory concentration (MIC) assay. The data are expressed as mean MIC values. MH, manuka honey; AH, artificial honey (sugars only).

## Role of $H_2O_2$ in the bactericidal effect of honey samples

Based on the ability of honey samples to generate $H_2O_2$ in 40% honey solution after 24 h of incubation, eight, the most antibacterially active, honey samples with a low (samples No. 1, 3, 6 and 8) and high capacity (7, 11, 15 and 27) to produce $H_2O_2$ were selected for further detailed analysis.

The bactericidal effects of eight honey samples were determined at their MIC value of 6% against *S. aureus* by counting the CFUs at different time points (Fig 3). As a control, $H_2O_2$ in 6% artificial honey solution was used at an MIC of 300 μM. The statistically significant reduction in CFU counts after 1 h of incubation was observed for six out of eight honey samples and the control sample containing $H_2O_2$. A significant decrease in CFU counts was found in all honey samples after 2 h of incubation compared to the positive control (6% artificial honey in cultivation media). In addition, the complete eradication of *S. aureus* was observed in samples No. 6 and 15 after 2 h of incubation and in all samples after 3 h of incubation. A similar significant bactericidal effect was also documented in the control experiment with a single dose of $H_2O_2$, with the complete eradication of *S. aureus* after 3 h of incubation.

To elucidate the role of $H_2O_2$ in inhibiting bacterial growth, 50% solutions of selected honey samples were treated with catalase for 2 h (Fig 4). The removal of $H_2O_2$ in catalase-treated samples resulted in a significant increase in MIC values, which ranged from 25% to 35% in all honey samples. However, the overall antibacterial effect after catalase treatment did not reduce to the level of artificial honey, with a mean MIC value of 42.5%.

Subsequently, we analysed the kinetics of $H_2O_2$ accumulation in eight honey samples at their MIC value of 6% (Fig 5). The maximum production of $H_2O_2$ in honey samples with a low capacity to produce $H_2O_2$ after 24 h was reached after 4 h of incubation at 37°C (Fig 5A). The maximum levels of $H_2O_2$ were in a range of 244 to 344 μM. In contrast honey samples

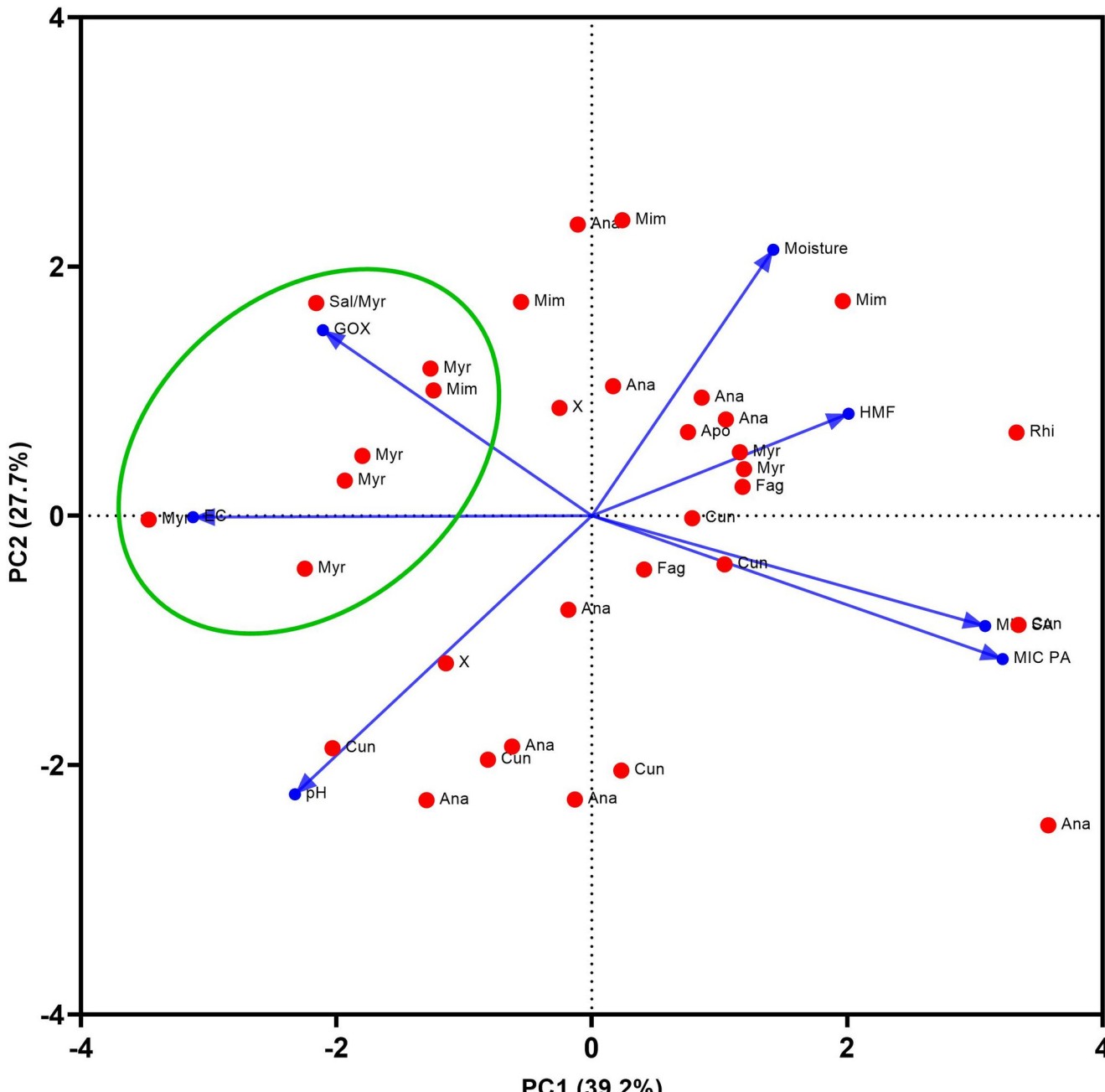

**Fig 2. Bioplot of the principal components analysis of honey from different botanical origins.** Ana, Anacardiaceae; Cun, Cunoniacea; Myr, Myrtacaea; Apo, Apocynaceae; Fag, Fagaceae, Sal, Salicaceae, Mim, Mimosaceae; X, multifloral honey type with no dominant pollen type. The green ellipse represents the area of the most honey samples belonging to the Myrtaceae family. Blue vectors represent the following variables: EC, electric conductivity; pH; Moisture; HMF, hydroxymethylfurfural; GOX, enzymatic activity of GOX; MIC SA, antibacterial activity against *Staphylococcus aureus*; MIC PA, antibacterial activity against *Pseudomonas aeruginosa*.

with a high capacity to produce $H_2O_2$ in 40% solution after 24 h accumulated the maximum level of $H_2O_2$ after 8 h of incubation or thereafter (Fig 5B). Although all eight honey solutions had the same concentration (6%), an obvious shift in the enzymatic kinetic profile of $H_2O_2$ production among the samples was observed.

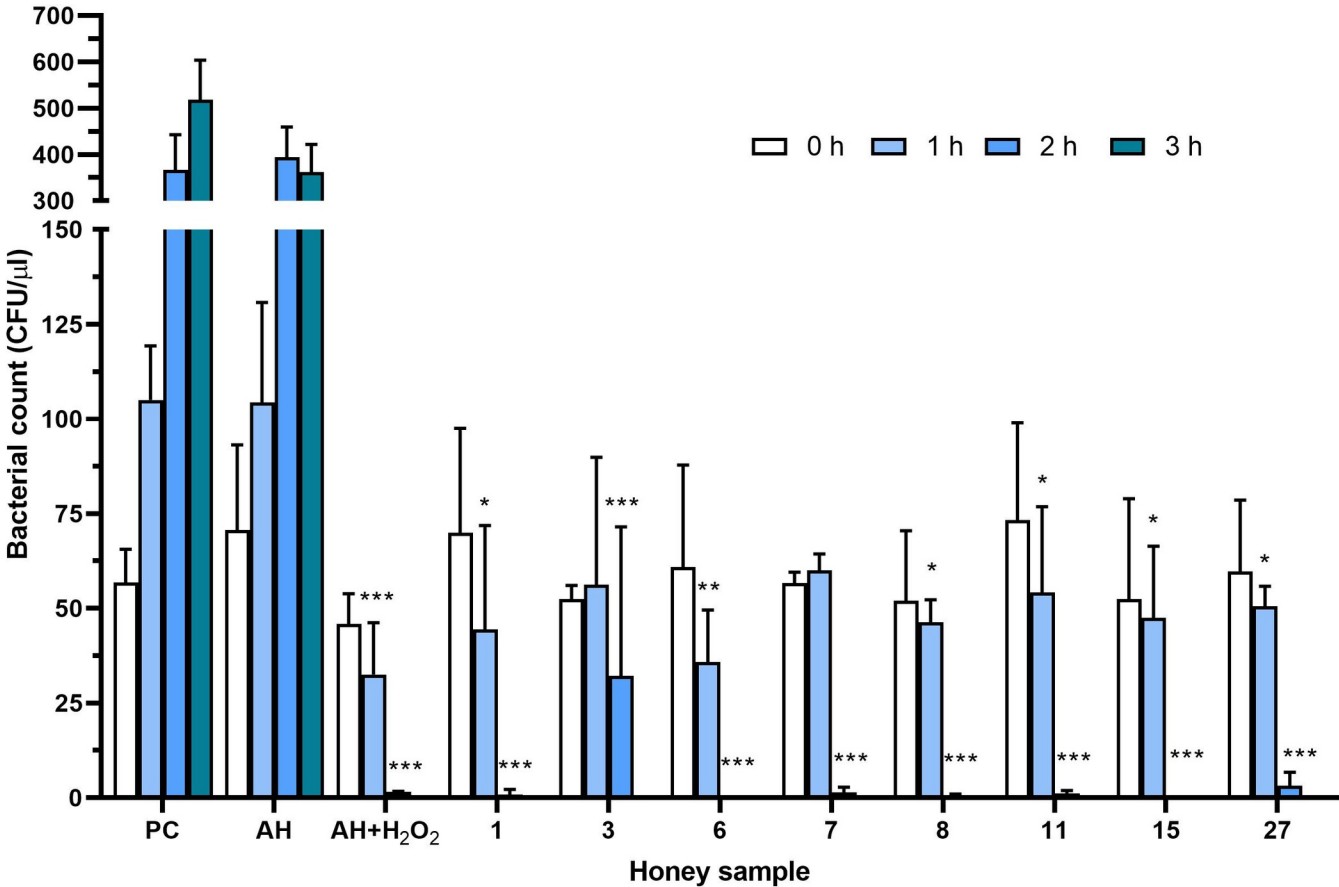

**Fig 3. Bactericidal effects of selected New Caledonian honey samples (n = 8) at their minimal inhibitory concentrations (MICs) at different time points against *Staphylococcus aureus*.** Bactericidal activity was determined by counting of colony-forming units (CFUs) and expressed as number of CFU/µL. PC, positive control (culture medium only); AH, artificial honey (sugars only). * $P < 0.05$, ** $P < 0.01$, *** $P < 0.001$.

The initial concentration of $H_2O_2$ in selected eight honey solutions at an MIC of 6% was below 100 µM (Fig 5), significantly lower than the used concentration of artificial added $H_2O_2$ (300 µM). We further determined the concentration of $H_2O_2$ in eight honey samples at their MICs (6%) and sub-MICs (3%) after 1, 2 and 3 h of incubation at 37°C (Fig 6). A marked increase in $H_2O_2$ production (above 150 µM) was observed in all honey samples at MIC after 1 h of incubation compared to samples at their sub-MIC values. Furthermore, none of the samples at sub-MIC was able to generate $H_2O_2$ at concentrations above 150 µM over the entire incubation period. Honey samples at MIC were able to significantly inhibit bacterial growth after 1 h of incubation (Fig 3), and a $H_2O_2$ concentration of 150 µM is the critical concentration for inhibiting the growth of *S. aureus* under study conditions.

### Effect of bacterial re-inoculation on honey antibacterial activity in an experimental setting

The subsequent experiments aimed to determine the antibacterial activity of selected honey samples and $H_2O_2$ after additional inoculation of *S. aureus*. After 4 h of incubation, each well containing either honey solution or $H_2O_2$ as antibacterial agent was re-inoculated with 10 µL of bacterial suspension at a concentration of $10^6$ CFU/mL. At these conditions, the MIC value of $H_2O_2$ was increased from 300 to 600 µM, and additional bacterial inoculum caused an

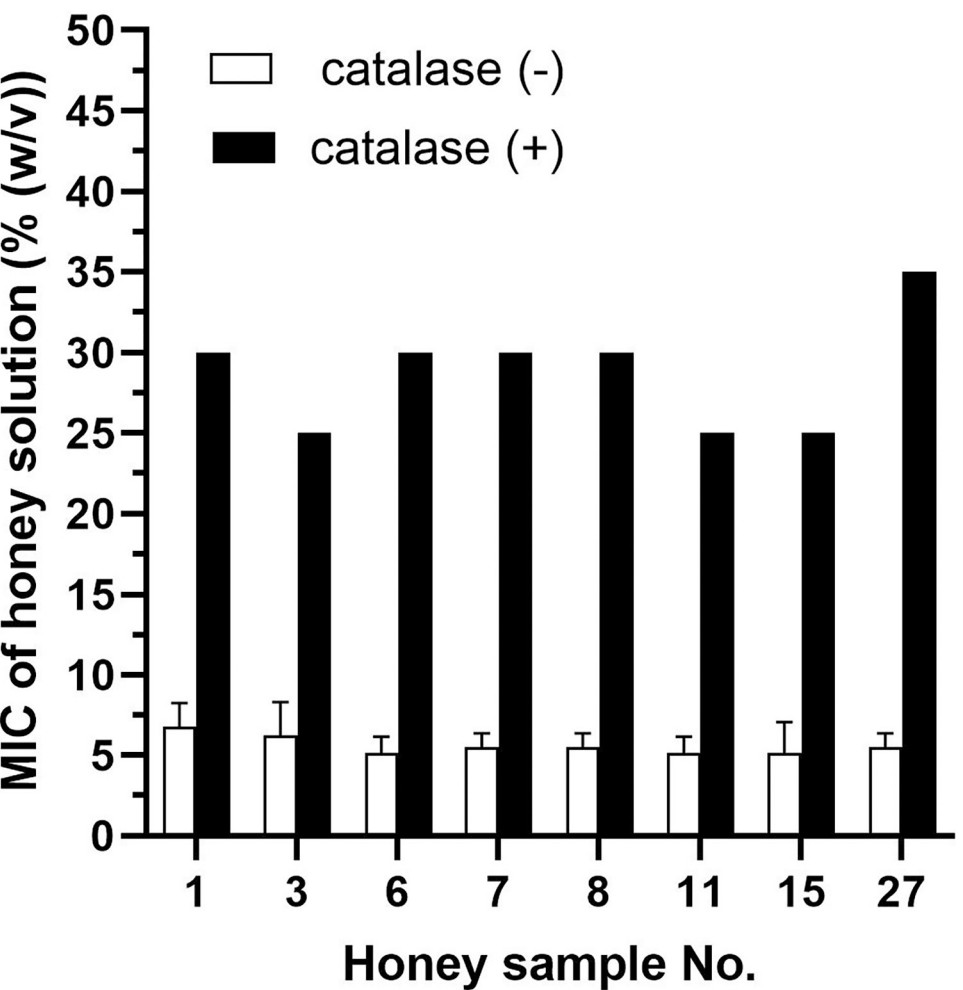

**Fig 4. Antibacterial activity of selected New Caledonian honey samples (n = 8) following catalase treatment against *Staphylococcus aureus*.** The antibacterial activity was determined with a minimum inhibitory concentration (MIC) assay. The data are expressed as mean MIC values.

increase in the MIC values of two honey samples, No. 1 and 3, by 25% and 40%, respectively (Fig 7). However, the MIC values of six out of eight honey samples remained unchanged.

## Discussion

The characterisation of the physicochemical and biological properties of honey samples from areas with a high floral diversity and endemicity is highly attractive for beekeepers, consumers as well as the pharmaceutical industry. Honey consumer behaviour, partially due to the Covid-19 pandemic, is increasingly associated with honey health benefits and the presence of organoleptic compounds [38]. The most common biological properties that consumers are aware of in honey are its natural antibacterial properties. It is widely accepted that manuka honey possesses therapeutic advantages over other types of honey because of its high antibacterial activity. However, we showed here and elsewhere [25] that certain types of honey, including honeydew honey, had similar or even superior antibacterial activity compared to manuka honey with a high content of MGO. Although the mechanism of action of all non-manuka honey types is $H_2O_2$-dependent, the honey polyphenol content may significantly contribute to

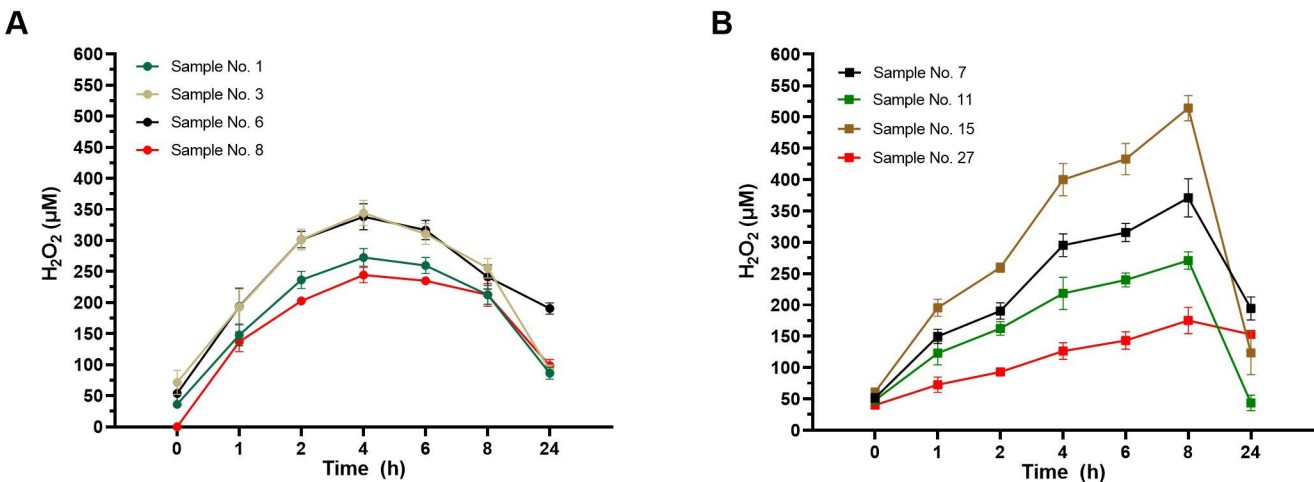

**Fig 5. Twenty-four-hour pattern of the generation of hydrogen peroxide ($H_2O_2$) in diluted honey samples.** The concentration of $H_2O_2$ was determined in selected honey samples at their MIC value of 6%. **A.** Honey samples No. 1, 3, 6 and 8, characterised by a low production of $H_2O_2$ at 40% dilution after 24 h. **B.** Honey samples No. 7, 11, 15 and 27, characterised by a high production of $H_2O_2$ at 40% dilution after 24 h. The data are expressed as mean concentration of $H_2O_2$.

honey antibacterial activity and is a key factor responsible for the augmented antibacterial effects of certain types of honey. The levels of $H_2O_2$ in honey samples rich in polyphenols is substantially higher compared to that of monofloral honey types [25, 39]. Yet, the exact mode of polyphenol action and their effect on honey antibacterial activity has not been described in detail.

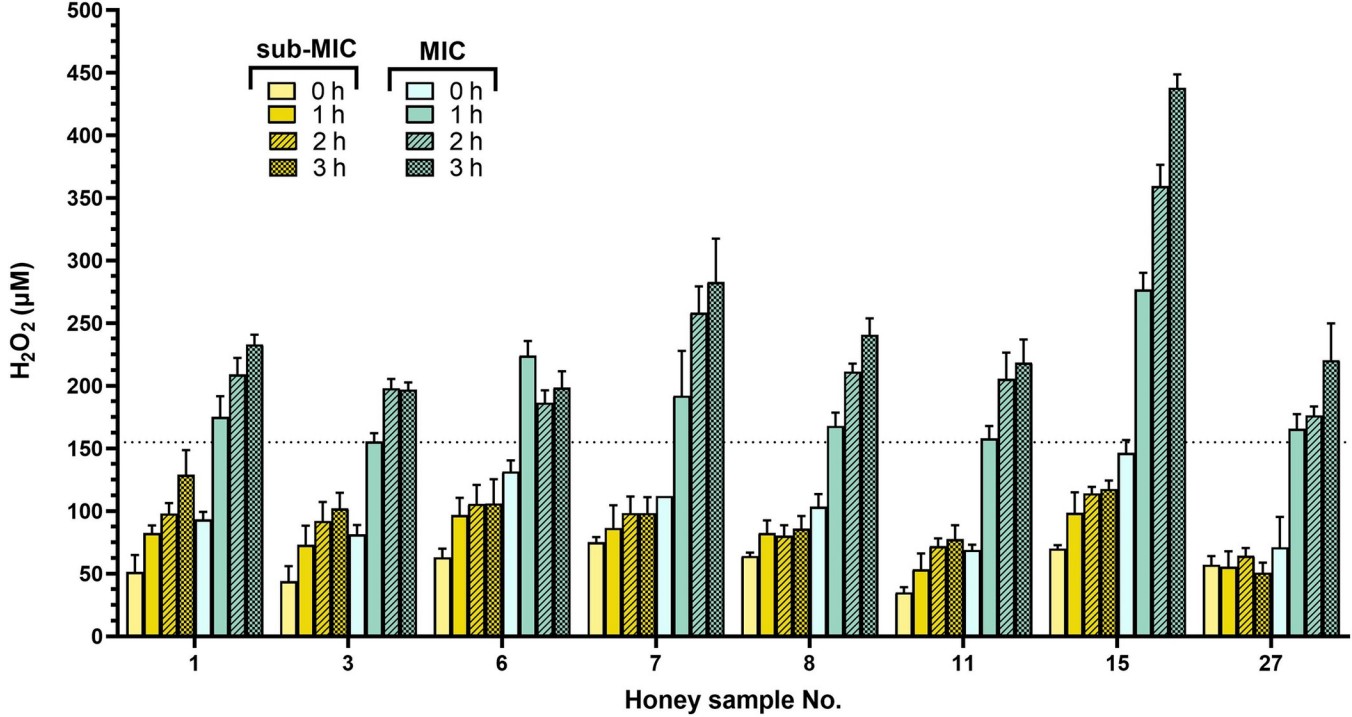

**Fig 6. Generation of hydrogen peroxide ($H_2O_2$) in diluted honey samples during 3 hours of incubation.** The concentration of $H_2O_2$ was determined in honey samples No. 1, 3, 6, 7, 8, 11, 15 and 27 at their MIC and sub-MIC values of 6% and 3%, respectively. The data are expressed as mean concentration of $H_2O_2$.

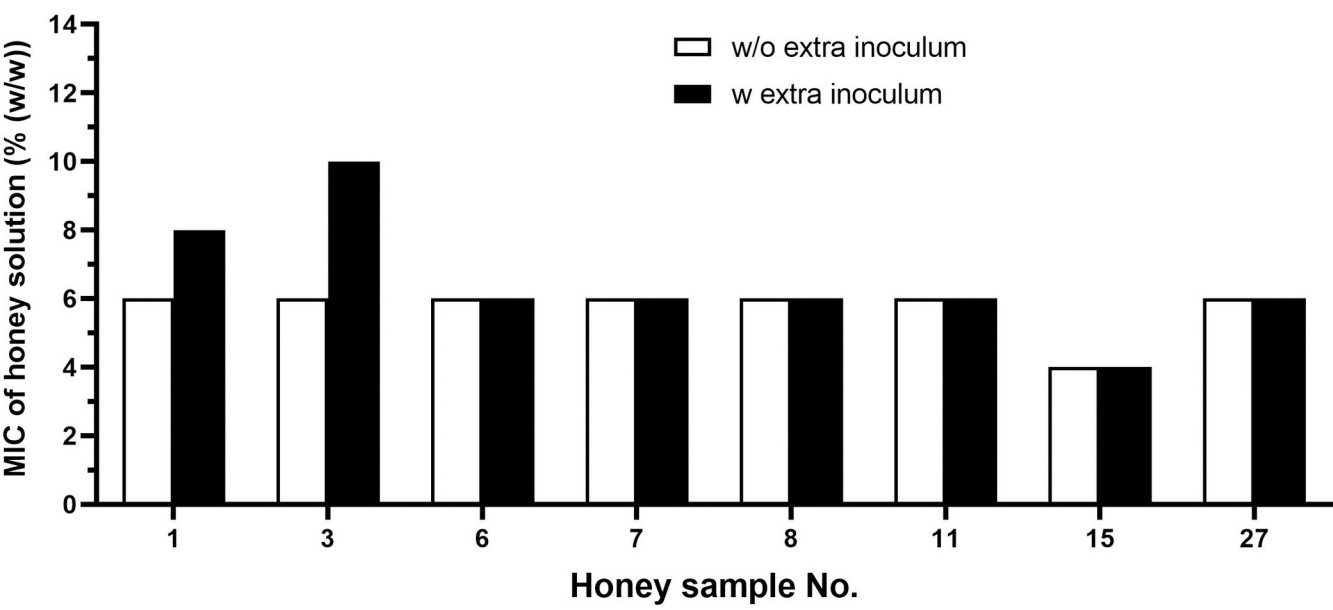

**Fig 7. Antibacterial activity of selected New Caledonian honey samples (n = 8) against *Staphylococcus aureus* after bacterial re-inoculation at 4 h.** The antibacterial activity was determined with a minimum inhibitory concentration (MIC) assay. The data are expressed as mean MIC values.

In this study, we clearly show that $H_2O_2$ is a key antibacterial compound in diluted New Caledonian honey samples. According to the PCA results, most of the honey samples containing pollen of the Myrtaceae family as the dominant pollen type were the most potent honey samples inhibiting bacterial growth. This honey type also had a high electrical conductivity Furthermore, these honey samples were also characterised by the highest values of GOX enzymatic activity when compared to the other honey botanical types. The level of accumulated $H_2O_2$ in 40% honey solutions after 24 h did not correlate with the enzymatic activity of GOX and the antibacterial activity of honey samples. In a previous study, the maximum accumulation of $H_2O_2$ occurred at 30%–50% honey dilution [40]. However, numerous studies characterising the production of $H_2O_2$ used honey diluted to 20%–30% strength [23, 28, 41–44]. Based on the results of this and of previous studies, the kinetic profile of $H_2O_2$ generation is not uniform in honey samples, and the maximum point of $H_2O_2$ production depends on the honey dilution rate and on various factors such as the concentrations of GOX enzyme, pollen-derived catalase and specific phytochemicals which enhance the enzymatic reaction [27]. The final concentrations of $H_2O_2$ can therefore vary in different honeys based on floral sources.

As mentioned above, the most potent honey samples inhibiting bacterial growth mostly contained pollen from the family Myrtaceae, which contains at least 133 genera and more than 3,800 species [45]. The family Myrtaceae is divided into two subfamilies, the capsular Leptospermoideae and the fleshy-fruited Myrtoideae. One of the most pronounced species of this family is *Leptospermum scoparium*, which is the botanical source of the well-known manuka honey, with different mechanisms of antibacterial action depending on the MGO level [46, 47]. Based on the enzymatic activity of GOX and the production of $H_2O_2$, supported by the results of the catalase honey treatment, we can conclude that the antibacterial effect of none of the analysed honey samples was mediated through the action of MGO.

Antibacterial effect of honey can also be attributed to action of other parameters such as bee-derived defensin-1 [48, 49], MRJP1 [50] and gluconic acid [51]. Defensin-1, an antibacterial peptide and MRJP1, the most abundant protein in honey, are expressed in hypopharyngeal bee glands and secreted during the processing of nectar into honey. A very recent proteomic

study showed no difference in the amount of MRJP1 between blossom and honeydew honeys [52]. Furthermore, honeys of different botanical and geographical origin might contain the comparable amounts of MRJP1 as well as other bee-secreted proteins/peptides and therefore differences in antibacterial efficacy of various honey samples are attributed to other antibacterial parameter(s) such as $H_2O_2$ [52]. Despite of the great difference in TPC values among New Caledonian honey samples, the SDS-PAGE electrophoretic profile of honey samples was comparable, indicating MRJP1 as a dominant protein.

To the best of our knowledge, no study has, so far, determined the critical concentration limit of $H_2O_2$ in honey solutions that is required for a sufficient inhibitory activity. Under laboratory conditions, most of the selected honey samples at MICs were able to reduce CFU counts after 1 h of incubation. The critical concentration of $H_2O_2$ in honey solutions after 1 h of incubation was above 150 μM, significantly inhibiting bacterial growth. However, a single dose of $H_2O_2$ in artificial honey at a concentration of 150 μM did not inhibit the growth of *S. aureus*, suggesting that phytochemicals play an important role in generating more toxic reactive oxidative species (ROS). The co-occurrence of $H_2O_2$ and phytochemicals found in honey stimulate the generation of long-lived and more toxic ROS compared to $H_2O_2$ alone [53]. Similarly, Masoura et al. (2020) suggested that the $H_2O_2$ antibacterial activity of natural honeys is a determinant of the ROS-inducing effect [51]. The authors speculated that darker honeys stimulate higher ROS generation compared to light ones due to the abundance of polyphenols in dark honeys [51].

From a clinical point of view, honey is most often used undiluted in wound care, and its further dilution with wound exudates depends on the wound type and the exudate level. Moderately and heavily exudating wounds can produce up to 5 mL of exudate per 10 cm$^2$ per 24 h [54], and it is important to avoid honey dilution to a concentration below that which gives the maximal rate of $H_2O_2$ production. Under *in vitro* conditions, the antibacterial efficacy of honey is determined by a standard procedure with a starting inoculum, which obviously does not correspond with wound conditions, where multiple bacterial species at different loads can be found. In this study, we investigated the capability of honey solutions at MIC values to counteract additional bacterial loads (extra inoculum) added after 4 h of incubation. The accumulated $H_2O_2$ in most of the honey solutions at their MIC values could inhibit the growth of additionally inoculated *S. aureus*, and the MIC values of the honey solutions remained unchanged. In contrast, the MIC value of $H_2O_2$ was doubled, from 300 to 600 μM, after the inoculation of additional bacteria, suggesting the importance of the continual slow release of $H_2O_2$ in diluted honey. A slow release of $H_2O_2$, at low concentrations, may prevent bacterial infections in wounds, and $H_2O_2$ at low concentrations is more stable. In another study, a continuous and slow release of $H_2O_2$ in diluted honey allowed the development of a hydrogel to achieve a continuous, controlled release of $H_2O_2$ for up to 72 h [55]. Also, $H_2O_2$ micro-encapsulated hydrogels demonstrated broad-spectrum antimicrobial activity with as little as 10 min of contact time [55].

According to a previous study, $H_2O_2$, at micromolar concentrations, exhibited wound healing properties by inducing vascular endothelial growth factor expression in human keratinocytes [56] and facilitating wound angiogenesis [57]. Low levels of $H_2O_2$ facilitated the regulation of wound inflammation [58]. Thus, $H_2O_2$, at low concentrations, acts as an antibacterial agent with wound healing properties, supporting the use of honey throughout the entire wound healing process.

## Conclusions

Analysis of New Caledonian honey samples revealed their strong antibacterial properties, primarily based on the inhibitory action of the generated $H_2O_2$. The ability of $H_2O_2$ production

in diluted honeys was not uniform among the samples. The critical concentration of $H_2O_2$ in diluted honeys responsible for inhibiting the growth of *S. aureus* was 150 μM. On the other hand, $H_2O_2$ itself at a concentration of 150 μM in artificial honey solution did not show any inhibitory activity, implying the important role of honey phytochemicals in the antibacterial effect of natural honey. Further research is needed to identify the honey phytochemicals which participate in strong antibacterial effects of New Caledonian honeys. Based on our findings, New Caledonian honeys represents a suitable source for medical-grade honey.

## Supporting information

**S1 Fig. Detailed representative analysis of SDS-PAGE protein profiles of collected New Caledonian honey samples (n = 33).**
(TIF)

**S1 Table. Qualitative pollen analysis of collected New Caledonian honey samples (n = 33).**
(PDF)

**S1 Raw images. Raw SDS-PAGE gels.**
(PDF)

## Acknowledgments

We thank New Caledonian beekeepers for providing honey samples.

## Author Contributions

**Conceptualization:** Marcela Bucekova, Romain Gueyte.

**Data curation:** Marcela Bucekova, Jana Godocikova, Romain Gueyte, Céline Chambrey.

**Formal analysis:** Marcela Bucekova, Jana Godocikova, Romain Gueyte, Céline Chambrey, Juraj Majtan.

**Funding acquisition:** Juraj Majtan.

**Investigation:** Marcela Bucekova, Jana Godocikova.

**Resources:** Juraj Majtan.

**Writing – original draft:** Marcela Bucekova, Jana Godocikova, Romain Gueyte, Céline Chambrey, Juraj Majtan.

**Writing – review & editing:** Marcela Bucekova, Jana Godocikova, Romain Gueyte, Céline Chambrey, Juraj Majtan.

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
