## [Decision Letter · Decision Letter 0]

12 Sep 2023

PONE-D-23-23826Characterisation of physicochemical parameters and antibacterial properties of New Caledonian honeysPLOS ONE

Dear Dr. Majtan,

Thank you for submitting your manuscript to PLOS ONE. After careful consideration, we feel that it has merit but does not fully meet PLOS ONE’s publication criteria as it currently stands. Therefore, we invite you to submit a revised version of the manuscript that addresses the points raised during the review process.

We look forward to receiving your revised manuscript.

Kind regards,

Filippo Giarratana

Academic Editor

PLOS ONE

“We thank New Caledonian beekeepers for providing honey samples. This work was supported by the Scientific Grant Agency of the Ministry of Education of the Slovak Republic and the Slovak Academy of Sciences VEGA 2/0022/22 and the Slovak Research and Development Agency under Contract No. APVV-21-0262.”

“This work was supported by the Scientific Grant Agency of the Ministry of Education of the Slovak Republic and the Slovak Academy of Sciences VEGA 2/0022/22 and the Slovak Research and Development Agency under Contract No. APVV-21-0262.

4. We note that Figure 1 in your submission contain map image which may be copyrighted. All PLOS content is published under the Creative Commons Attribution License (CC BY 4.0), which means that the manuscript, images, and Supporting Information files will be freely available online, and any third party is permitted to access, download, copy, distribute, and use these materials in any way, even commercially, with proper attribution. For these reasons, we cannot publish previously copyrighted maps or satellite images created using proprietary data, such as Google software (Google Maps, Street View, and Earth). For more information, see our copyright guidelines: http://journals.plos.org/plosone/s/licenses-and-copyright.

Additional Editor Comments:

Dear Author,

please improve the paper with all the Reviewer's suggestions/requirements.

Reviewers' comments:

Reviewer's Responses to Questions

**Comments to the Author**

1. Is the manuscript technically sound, and do the data support the conclusions?

Reviewer #1: Partly

Reviewer #2: Yes

2. Has the statistical analysis been performed appropriately and rigorously? 

Reviewer #1: I Don't Know

Reviewer #2: Yes

3. Have the authors made all data underlying the findings in their manuscript fully available?

Reviewer #1: No

Reviewer #2: Yes

4. Is the manuscript presented in an intelligible fashion and written in standard English?

Reviewer #1: Yes

Reviewer #2: Yes

5. Review Comments to the Author

Reviewer #1: The manuscript entitled “Characterisation of physicochemical parameters and antibacterial properties of New Caledonian honeys” investigates the antibacterial activity of different honey samples in light of the specific hydrogen peroxide content. The manuscript seems well written, with the English requiring only minor adjustments (considering my knowledge). The topic is interesting and worthy of attention but as it stands the article has some shortcomings that do not allow it to be published.

In detail:

1. There is not a clear link between the introduction and the goals of the study.

2. The MMs are not clear on several points and need to be improved especially as regards the antimicrobial evaluations.

3. The results section shows data for which the methods of analysis have not been described (see subsection 3.1.).

4. The Results section reports information that looks more like discussion than results. This last is repeated several times so I would suggest a massive editing of the article to remove discussions from the result section.

Therefore, I suggest either rejecting the article or major revisions.

Abstract

In my opinion, the abstract does not elucidate well what the authors did during the study. The materials and methods section is not fully explanatory. The authors used abbreviations that may not be understood by a non-addicted reader. I suggest editing the abstract summarizing better all the sections of the study.

Introduction

The introduction is not badly written, but the authors should be focused better the aims of the study which are several and, in my opinion, are not well linked with the information reported in the introduction. I would suggest critically reviewing the introduction.

Line 51: Use another word instead of “consumption”.

Line 57: Improve the connection between this sentence and the periods before and after.

Lines 63-64: I would suggest moving this sentence before the previous period.

Line 70 and 72: Add references.

Line 82: add a reference.

Line 90: it is not clear what the authors mean by “uncover”. Please, improve.

Lines 90-91: It is not clear the connection between this sentence and the previous sentence. Could the author explain better, please?

Line 93: What do the authors mean by “high endemicity”? Please, improve and explain better.

Line 99: Why did the authors investigate the activity of glucose oxidase but they never discuss it in the introduction?

Line 101: What do the authors mean by "honey diluted? Please, improve.

Materials and methods

Lines 107-108: improved this sentence, it seems that the period is missing a final part.

Line 110: please, somewhere describe what UMF means.

Line 112: immediately before or after what?

Line 119: in my opinion, this paragraph does not make sense reported in this way. Improve scribing also how the samples were stored and prepared for the antibacterial activity. Furthermore, report the manufacturers where the bacteria were acquired.

Lines 120: isolates instead of isolate.

Line 125: what does “by laboratories” mean? Improve

Line 124: please, improve this paragraph by reporting at least a brief description of the methods used (“Harmonized methods of the internationals honey commission”) and the manufacturers of the instruments used for the different parameters’ determination.

Lines 159-161: it is unclear what and how was determined. The mic of what versus whom? diluted in broth how? Why? In what quantity? Improve.

Line 170: it is difficult for the reader to follow the article. The method is not well described. Why MHB “or” “diluted honey”? It seems that a part of the method is missing. Improve.

Line 172: visually in which way? Turbidity? Improve.

Lines 178-181: This section should be moved above before or inserted together with the description of the method adopted. It is not clear the difference between MIC and MBC.

Line 184: what do the authors mean by spotted?

Results

Line 206: This analysis was not defined in the MM. Add the methods used in the MM. Improve.

Line 225 legislative criterium of New Caledonia? Or another country/continent?

Lines 227-288: this looks more like discussions than results.

Lines 234-235: this looks more like discussions than results.

Please, rewrite the article so that only results and not discussions are reported in the results section. A it is, it is difficult to follow the results obtained by the analyses.

Discussion

The discussions seem well written and discretely discuss the results obtained. However, considering the need for major improvements in the results and the materials and methods sections, the discussions deserve further revision in the light of future updates by the authors.

Figure and tables

The captions of both figures and tables are fully explanatory and they are both well presented.

Reviewer #2: The work completed on the topic of Characterisation of physicochemical parameters and antibacterial properties of New Caledonian honeys is very applied and conducted in details.

The results and their critical discussions are stronger point for this research article.

Manuscript would be very interesting, if it had some pictures of New Caledonian Honeys samples used in experiments, but that is not necessary

I suggest to use selected type of Caledonian honeys, which are rich in Bioactivities for innovative health care products.

Manuka Honey is sold very expensive in Health shops, because there is no competition with other health grade honeys in the market,

I hope Caledonian honeys would be available in market at a cheaper cost for people to afford the cost, please explore such possibility.

6. PLOS authors have the option to publish the peer review history of their article (what does this mean?). If published, this will include your full peer review and any attached files.

Reviewer #1: No

Reviewer #2: No

---

## [Author Response · Author response to Decision Letter 0]

18 Sep 2023

Response to reviewers’ comments and suggestions

Reviewer #1

The manuscript entitled “Characterisation of physicochemical parameters and antibacterial properties of New Caledonian honeys” investigates the antibacterial activity of different honey samples in light of the specific hydrogen peroxide content. The manuscript seems well written, with the English requiring only minor adjustments (considering my knowledge). The topic is interesting and worthy of attention but as it stands the article has some shortcomings that do not allow it to be published.

In detail:

1. There is not a clear link between the introduction and the goals of the study.

Thank you for comment. We rewrote the introduction and added more background information which are in direct connection with the aims of our study. Please see our revised ms and answers for your raised issues below.

2. The MMs are not clear on several points and need to be improved especially as regards the antimicrobial evaluations.

Thank you for your valuable comments and suggestions. We agree that some details were missing and also some statement could be confusing. Therefore, we improved the description of methods, especially the determination of antibacterial activity of honey. Please see bellow responses for your comments/suggestions. Please see also our revised ms.

3. The results section shows data for which the methods of analysis have not been described (see subsection 3.1.).

The pollen analysis was used for determining of floral origin of honey. It was mentioned in the first paragraph in MM (paragraph “Honey samples“): „The samples were harvested in 2021 and identification of the floral source of each honey sample using mellisopalynological analysis according to Erdtman [32] was performed by CARI asbl, Louvain-La-Neuve, Belgium” However, we added the reference supporting this analysis. Please see our revised ms.

Pollen analysis of honey or mellisopalynoligcal analysis is a gold standard determining honey botanical type. This analysis very much depends on broad experience and only a few commercial laboratories are able to do it. Therefore, pollen analysis in our case was provided as a commercial service. 

4. The Results section reports information that looks more like discussion than results. This last is repeated several times so I would suggest a massive editing of the article to remove discussions from the result section.

Thank you for your suggestion. We removed all sentences/paragraphs which were related with discussion. On the other hand, we described obtained results in detail what is very important for researchers/readers who are not familiar with microbiological techniques and honey research. In addition, all references were removed from results section. We believe that the results section is improved but keeps enough information for readers. Please see our revised ms.

In addition, according to PLos One guidelines, figure legends should be placed in manuscript where the particular Figure is mentioned. Therefore, it looks a bit messy and too wordy.

Abstract

In my opinion, the abstract does not elucidate well what the authors did during the study. The materials and methods section is not fully explanatory. The authors used abbreviations that may not be understood by a non-addicted reader. I suggest editing the abstract summarizing better all the sections of the study.

Thank you for your comment and suggestion. We agree that section MM was omitted in abstract. Therefore, we revised and rewrote the abstract taking into account the final word limit for abstract (300 words). All abbreviations were shown in full in abstract and some of them were removed.

Introduction

The introduction is not badly written, but the authors should be focused better the aims of the study which are several and, in my opinion, are not well linked with the information reported in the introduction. I would suggest critically reviewing the introduction.

Thank you for your valuable comment and suggestion. We rewrote some parts of introduction (also including your suggestions bellow) and we added more information about hydrogen peroxide, its production and the role in antibacterial activity of honey. We believe that revised introduction linked with the goals of the study.

Line 51: Use another word instead of “consumption”.

Thank you. It was reworded. Please see revised ms.

Line 57: Improve the connection between this sentence and the periods before and after.

Thank you for suggestion. We added the sentence to improve connection between two paragraphs.

Lines 63-64: I would suggest moving this sentence before the previous period.

Thank you for suggestion. We moved the sentence at the of previous paragraph.

Line 70 and 72: Add references.

Selected and most appropriate references were added.

Line 82: add a reference.

Reference was added.

Line 90: it is not clear what the authors mean by “uncover”. Please, improve.

Thank you. It was reworded. Please see revised ms.

Lines 90-91: It is not clear the connection between this sentence and the previous sentence. Could the author explain better, please?

Thank you for comment. We improved the clarity and connection between the sentences. Please see revised ms.

Line 93: What do the authors mean by “high endemicity”? Please, improve and explain better.

Thank you for your comment. Term “high endemicity” was used for plants in New Caledonia. We modified the sentence to improve the clarity. Please see our revised ms.

Line 99: Why did the authors investigate the activity of glucose oxidase but they never discuss it in the introduction?

Thank you for your comment. We added this information in introduction and highlighted the role of glucose oxidase in honey antibacterial activity through the production of hydrogen peroxide. Please, see our revised ms.

Line 101: What do the authors mean by "honey diluted? Please, improve.

Thank you for your comment. We aimed to investigate antibacterial effect of honey against S. aureus at condition in which honey was diluted to its minimal inhibitory concentration against S. aureus. We improved the clarity of the sentence. Please see our revised ms.

Materials and methods

Lines 107-108: improved this sentence, it seems that the period is missing a final part.

The sentence was improved. Please see our revised ms.

Line 110: please, somewhere describe what UMF means.

A shortcut “UMF” has been shown in full meaning in the introduction as appeared for the first time in manuscript.

Line 112: immediately before or after what?

The sentence clarity was improved. See our revised ms.

Line 119: in my opinion, this paragraph does not make sense reported in this way. Improve scribing also how the samples were stored and prepared for the antibacterial activity. Furthermore, report the manufacturers where the bacteria were acquired.

Thank you for suggestion. We added more information about the bacterial cultures, their origin, storing and working conditions. Please see our revised ms.

Lines 120: isolates instead of isolate.

It was corrected.

Line 125: what does “by laboratories” mean? Improve

Thank you for your comment. We corrected it.

Line 124: please, improve this paragraph by reporting at least a brief description of the methods used (“Harmonized methods of the internationals honey commission”) and the manufacturers of the instruments used for the different parameters’ determination.

As it was mentioned in manuscript, physicochemical analyses were carried out at Cari asbl (Belgium). The methods used for these analyses were accredited methods; however, we do not have information about the manufacturers of used laboratory machines/devices. This was done as a commercial service. We believe that description of the standard methods for basic physicochemical parameters of honey is not necessary since they are widely used in Europe. 

Lines 159-161: it is unclear what and how was determined. The mic of what versus whom? diluted in broth how? Why? In what quantity? Improve.

Thank you for comment. We improved the clarity of the paragraph to be clearer for readers. Please see our revised ms.

Line 170: it is difficult for the reader to follow the article. The method is not well described. Why MHB “or” “diluted honey”? It seems that a part of the method is missing. Improve.

Thank you for your comment. We improved the clarify and description of the method. We added information about positive and negative control. Description about dilution of honey samples was shown at the bottom of the same paragraph. This method has been widely published and also used numbers of honey researchers. It is a standard method but accommodated to honey.

Line 172: visually in which way? Turbidity? Improve.

Thank you for your comment. Yes, the lack of visual turbidity was used to determine the minimal inhibitory concentration of honey samples. We improved the description of this method. In addition, based on Clinical and Laboratory Standards Institute (CLSI) guidelines, visual inspection of bacterial growth is a standard approach. 

Lines 178-181: This section should be moved above before or inserted together with the description of the method adopted. It is not clear the difference between MIC and MBC.

Thank you for your comment. We defined what MBC means and corrected the sentences and also used proper reference. MIC and MBC of honey represent common parameters characterizing the antibacterial activity of honey. MIC expresses inhibitory activity and MBC expresses bactericidal activity of particular honey samples. Please see our revised ms.

Line 184: what do the authors mean by spotted?

Thank you for your comment. Spotting means that the aliquots in a final volume of 10 ul were dropped on the agar plate. The drops let soak into agar media and plate was incubated overnight. It is common procedure in microbiology. There are hundreds of papers where same technique and its description was used.

Results

Line 206: This analysis was not defined in the MM. Add the methods used in the MM. Improve.

The pollen analysis – determining of floral origin of honey – was mentioned in the first paragraph in MM (paragraph “Honey samples“): „The samples were harvested in 2021 and identification of the floral source of each honey sample was performed by laboratories at CARI asbl, Louvain-La-Neuve, Belgium.” However, we added the reference supporting this analysis. It was provided as a commercial service. Please see our revised ms.

Line 225 legislative criterium of New Caledonia? Or another country/continent?

Thank you for comment. The legislative criteria according to Codex standards for honey. Thus, it is international criterium. We added this information into result section (without references that should not be used in Results section)

Lines 227-288: this looks more like discussions than results.

Thank you for comment. We removed the sentence from results section.

Lines 234-235: this looks more like discussions than results.

Thank you for comment. We removed the sentence from results section.

Please, rewrite the article so that only results and not discussions are reported in the results section. A it is, it is difficult to follow the results obtained by the analyses.

Thank you for your suggestion. We removed all sentences/paragaraphs which were related with discussion. On the other hand, we described obtained results in detail what is very important for researchers who are not familiar with microbiological techniques and honey research. In addition, all references were removed from results section. We believe that the results section is improved but keeps enough information for readers. Please see our revised ms.

Discussion

The discussions seem well written and discretely discuss the results obtained. However, considering the need for major improvements in the results and the materials and methods sections, the discussions deserve further revision in the light of future updates by the authors.

Thank you for your comment. As you indicated, discussion is well written. Therefore, we made some changes and add some more information to discussion section. However, no changes in data or in graphs were made. Therefore, we believe that discussion is robust, actual and taking into account all important results of our present study. Please see our revised ms.

Reviewer #2

The results and their critical discussions are stronger point for this research article. Manuscript would be very interesting, if it had some pictures of New Caledonian Honeys samples used in experiments, but that is not necessary.

Thank you for your suggestion. Unfortunately, pictures of New Caledonian honey samples would not be very attractive. All samples are in the same plastic containers and color of the honey samples are not very much diverse. In addition, we keep the samples at 4°C (in order to stabilise the biological properties) and most of the samples are in solid state. 

Manuka Honey is sold very expensive in Health shops, because there is no competition with other health grade honeys in the market. I hope Caledonian honeys would be available in market at a cheaper cost for people to afford the cost, please explore such possibility.

Thank you for your suggestion. Actually, overall antibacterial activity is comparable with manuka honey and in conclusion section as well as in abstract we advocate New Caledonian honey samples as a potential source for medical-grade honey or honey-based wound care products.

---

## [Editor Report · Decision Letter 1]

18 Oct 2023

Characterisation of physicochemical parameters and antibacterial properties of New Caledonian honeys

PONE-D-23-23826R1

We’re pleased to inform you that your manuscript has been judged scientifically suitable for publication and will be formally accepted for publication once it meets all outstanding technical requirements.

Kind regards,

Filippo Giarratana

Academic Editor

PLOS ONE
---

## [Editor Report · Acceptance letter]

23 Oct 2023

PONE-D-23-23826R1 

Characterisation of physicochemical parameters and antibacterial properties of New Caledonian honeys 

Dear Dr. Majtan:

I'm pleased to inform you that your manuscript has been deemed suitable for publication in PLOS ONE. Congratulations! Your manuscript is now with our production department. 

Kind regards, 

on behalf of

Dr. Filippo Giarratana 

Academic Editor

PLOS ONE